# Addressing Urban Sprawl from the Complexity Sciences

**Martí Bosch** [1,*] **, Jérôme Chenal** [1] **and Stéphane Joost** [1,2]

[1]  Urban and Regional Planning Community (CEAT), School of Architecture, Civil and Environmental Engineering (ENAC), École Polytechnique Fédérale de Lausanne (EPFL), 1015 Lausanne, Switzerland; jerome.chenal@epfl.ch (J.C.); stephane.joost@epfl.ch (S.J.)

[2]  Laboratory of Geographic Information Systems (LASIG), School of Architecture, Civil and Environmental Engineering (ENAC), École Polytechnique Fédérale de Lausanne (EPFL), 1015 Lausanne, Switzerland

*  Correspondence: marti.bosch@epfl.ch; Tel.: +41-21-69-34435

**Abstract:**   Urban sprawl is nowadays a pervasive topic that is subject of a contentious debate among planners and researchers, who still fail to reach consensual solutions. This paper reviews controversies of the sprawl debate and argues that they owe to a failure of the employed methods to appraise its complexity, especially the notion that urban form emerges from multiple overlapping interactions between households, firms and governmental bodies. To address such issues, this review focuses on recent approaches to study urban spatial dynamics from the perspective of the complexity sciences. Firstly, spatial metrics from landscape ecology provide means of quantifying urban sprawl in terms of increasing fragmentation and diversity of land use patches. Secondly, cellular automata and agent-based models suggest that the prevalence of urban sprawl and fragmentation at the urban fringe emerge from negative spatial interaction between residential agents, which seem accentuated as the agent's preferences become more heterogeneous. Then, the review turns to practical applications that employ such models to spatially inform urban planning and assess future scenarios. A concluding discussion summarizes potential contributions to the debate on urban sprawl as well as some epistemological implications.

**Keywords:** urban sprawl; complexity; land use change; landscape metrics; fractals; cellular automata; agent-based models; urban planning

## 1. Introduction

The world's population is undergoing an unprecedented trend of urbanization, with more than half of its population currently living in urban areas [1]. In the beginning of the 21st century, the amount of land occupied by cities roughly accounted for three percent of the world's arable land, but current prospects at decreasing urban densities forecast that it might rise to 5–7 percent by 2030 [2]. Although the numbers might still appear relatively small, the environmental footprint of cities has significant implications at the global scale, for their functioning produces 78% of the earth's greenhouse gases [3].

The increasing pervasiveness of urban sprawl has raised numerous sustainability concerns, mainly in terms of the loss of natural land and increased traffic-related emissions [4]. As a response, many planning efforts such as greenbelts, urban growth boundaries or land use zoning have been devoted to contend with urban sprawl, yet empirical evaluations of their success are scarce [5], especially since the impact of planning on the actual urban development is hard to disentangle from that of socioeconomic drivers and technological forces [6].

Paralleling the contentious debate on the desirability of urban sprawl, recent holistic perspectives to the study of complex systems have transformed the way in which forms and processes are understood within human and natural systems. Landscape ecologists have developed frameworks to study the relationships between the spatial geometry and composition of landscapes and the ecological processes that occur upon them [7], which are central to anticipating the effects of increasing land conversion due to urbanization [8]. Concurrently, computational approaches such as cellular automata and agent-based models have provided novel insights into the mechanisms that generate the complex forms of contemporary cities [9], which exhibit scaling relationships similar to those observed within a wide diversity of biological organisms [10]. The aim of the present article is to review potential ways in which such approaches can enlighten the sprawl debate, urban theory and urban planning.

The paper begins by reviewing the literature that examines urban sprawl from the viewpoint of urban economics and the regional sciences, which has been the prevalent approach among the early studies of urban sprawl. This perspective quantifies the morphology of urban sprawl by means of a set of indices, which are usually constructed based on of the distribution of population, housing and employment over the various districts and counties that configure a metropolitan area [11]. In order to assess the impacts of urban sprawl, such studies often explore the empirical correlations between the foregoing indices and a variety of indicators of travel behavior, environmental performance and public health. The evidence reviewed in the present article suggests that the deficiencies of the urban economics and regional sciences approach to urban sprawl are threefold. Firstly, the definitions of urban sprawl remain elusive, which impedes establishing consensual methods to measure it. Secondly, literature findings on the impacts of urban sprawl on transportation and the environment are equivocal, mainly because the employed sprawl indices do not reflect the complexity and diversity of urban forms of contemporary cities. However, most importantly, many studies misappreciate the extent to which the interactions between households, firms, governments and other agents shape contemporary cities, often leading to outcomes that had not been directly included within the designed plans [12].

The review then turns to studies that address urban form from the perspective of the complexity sciences. The term *complexity* refers to "the high-order phenomena arising from a system's many connected, interacting subcomponents" [13] (p. 281) and is concerned by "both dynamics (i.e., processes) and structure (i.e., patterns and configurations)". Like traditional sciences, the *complexity sciences* are concerned with the study of regularities. Nevertheless, unlike traditional sciences, the complexity sciences do not focus on simple cause–effect relationships but on exploring how generative rules can lead to complex behaviors that resemble real world phenomena [14]. Based on the above definitions, two main groups of studies have been reviewed. On the one hand, empirical studies that employ spatial metrics from landscape ecology suggest that contemporary cities show a global tendency towards fragmented landscapes that match many connotations of urban sprawl. On the other hand, cellular automata and agent-based simulations suggest that the observed fragmentation emerges from residential preferences for suburban environments and heterogeneity between agents.

Subsequently, the review shifts to practical applications that exploit approaches from the complexity sciences to spatially inform urban planning and assess future scenarios. A concluding discussion reviews implications for the sprawl debate and urban theory.

## 2. Urban Sprawl from the Perspective of Urban Economics and Regional Sciences

### 2.1. Defining and Quantifying Sprawl

The first reference to the term urban sprawl was made by Earle Draper, as part of a conference of urban planners of the southeastern United States in 1937. The topic acquired striking relevance during the second half of the twentieth century, and, ever since then, it has been continuously spreading to a wide range of domains. In one of the early efforts to characterize urban sprawl, Harvey and Clark [15] criticized the lack of accepted definitions of the term and delineated three physical patterns of sprawl, namely continuous low density, ribbon development and leapfrog development. However, besides a

set of archetypes, "sprawl is a matter of degree" [16] (p. 520). For instance, to what extent polycentric urban forms might be considered sprawl is not clear [17]. On the other hand, sprawl has also been associated with a dysfunctional spatial segregation of land uses [18]. Despite remarkable efforts to assemble different acceptations of sprawl [19], the research community still fails to agree on a common definition of urban sprawl, especially since the term can be heard from very diverse practitioners and its interpretation is likely to depend on the discipline and the context of application. Be that as it may, some prominent characteristics of sprawl reappear often in the literature, such as scattered development, low density, decentralization to the urban periphery, segregation of land uses and low accessibility, which often results in automobile dependence.

From the semantic ambiguity of urban sprawl follows a lack of consensual methods to measure it. In consonance with the preceding traits, the prevalent approach is to treat sprawl as a multidimensional phenomena, and several dimensional decompositions have been proposed throughout the literature. Some of them are rather simple, such as the four dimensions proposed by Ewing et al. [20], i.e., density, land use mix, centering and accessibility, whereas Galster et al. [19] unravel sprawl further and identify eight dimensions, i.e., density, continuity, concentration, clustering, centrality, nuclearity, land use mix and proximity. Nevertheless, such decompositions seem fuzzy, even for the simplest cases, e.g., the accessibility to a given facility is related to the density, the number of activity centers and the land use mix. In point of fact, when computing the dimensions of sprawl for the largest metropolitan areas of the United States, concentration, proximity clustering and centrality are significantly correlated in Galster et al. [19], and the same holds for density and connectivity in Ewing et al. [20]. When calculating aggregate indices, such intricateness might result in overemphasizing certain features of sprawl. In the discussion between different views towards urban sprawl of Ewing et al. [21], some authors argue that many of the research results rely heavily on how the sprawl indices are constructed, which is "highly subjective and depends upon very specific and not necessarily generally accepted definitions of sprawl" (p. 15). As a matter of fact, the loose manner in which sprawl indices can be formulated has led to wildly different ratings of sprawl given to different metropolitan areas by different analysts [22].

## *2.2. The Costs of Sprawl and Contradicting Evidence*

While the multiplicity of perspectives adopted to investigate urban sprawl highlights the relevance of the topic, the involved ambiguity paves the way for incomplete assessments, endogenous biases and premature claims, which are often accused to be politically motivated. The vast report of the Real Estate Research Corporation [23] has been a noteworthy source of controversy. For instance, Altshuler et al. [24] stated that it includes few rigorous calculations on the car use decrease, while Windsor [25] concluded that the claimed energy savings in their alternative scenarios are more a result of their assumptions rather than of the density. Also as response to the report, Gordon and Wong [17] pointed to the evidence of reduced trip lengths of the suburban residents of large polycentric cities to suggest that, as cities grow, travel demands are accommodated through decentralization of the employment centers. More broadly, Haines [26] determined that studies that consider only centralization and sprawl resolve that centralization is the most energy-efficient option, whereas studies that additionally consider polycentric urban forms favor the latter. Similar controversies arose from the global strong correlations between density and gasoline consumption established empirically by Newman and Kenworthy [27], mainly because density alone neglected the complexity and diversity of the analyzed urban patterns. Overall, in a thorough review of empirical studies, Hall [28] discerned that literature findings relating transportation and urban form when compared with each other are equivocal, and resolved that travel is globally more linked to income than density.

Although the sprawl debate has focused more on transportation issues, additional environmental implications of urban form require careful consideration. Urban areas exert significant influence on its surrounding ecosystems and the services that they provide to humans and other living beings [29]. While many studies have found empirical correlations between environmental performance and

aggregate measures of urbanization such as density or the extent of the built-up area, such measures do not reflect the complexity and diversity of existing urban patterns, and are strongly affected by the different definitions of city boundaries [30]. For instance, the comparative study of five UK cities of Tratalos et al. [31] associated higher density to poorer urban biodiversity and environmental services such as carbon sequestration, storm water interception or alleviation of maximum temperatures, yet sites with similar densities show substantial variability on the environmental performance, even within the comparable conditions among the UK. Overall, the effects of urban density and form are hard to isolate since their impact on environmental performance is mediated by the local environmental characteristics of the site.

### 2.3. Managing Urban Sprawl and Self-Organization

While the connection between urban sprawl and urban growth appears obvious, sprawl is very often assessed in a cross-sectional matter. The first to attest to this deficiency were Harvey and Clark [15], alleging that "sprawl is a form of growth" (p. 6), and should therefore be assessed with a reasonable time span. Otherwise, the development costs might be exaggerated, since initial sprawl might be a first step towards posterior densification. For instance, the empirical regression of Peiser [32] suggested that, in the absence of zoning regulations and in a competitive land market, initial discontinuous development will be followed by later infill, resulting in higher densities.

Although growth management policies such as land use zoning and the adoption of urban growth boundaries have significant impacts on urban form and density, uncoordinated implementations might result in shifting the urban development to other neighboring communities, therefore increasing urban sprawl at the scale of the metropolitan area as a whole [33]. For example, in a long-term study of the Seattle region, Robinson et al. [34] found that, while growth management efforts lead to an increased housing density within the existing city limits, they inadvertently encouraged sprawl outside the designated growth boundaries. On the other hand, the implementation of comprehensive land use plans also confront notable difficulties. For instance, the conformance evaluation of the Israel's Central District land use plan of Alfasi et al. [35] found fundamental gaps between the original land use assignments and the actual development, mainly attributed to a succession of local amendments to the plan performed gradually on a case-by-case basis. Similar conclusions were attained by Abrantes et al. [36] in an implementation evaluation of the land use plans of Lisbon's metropolitan region, where most noncompliant urban development occurred at the expense of reducing and fragmenting agricultural land.

As highlighted by many early studies of sprawl, buyer preferences for suburban environments are among its main drivers. Nonetheless, household location choices also ponder other criteria such as accessibility to work and shopping centers [37]. Resuming their former findings of transportation and decentralization, Gordon et al. [38] disclosed that commuting times of the 20 largest American metropolitan areas remained stable despite rapid suburban growth, which, following investigations by Levinson and Kumar [39], attributed to evidence of an increased share of trips with both origin and destination in the suburbs. The authors suggested that households and job opportunities mutually co-located to optimize costs and travel times, resulting in the polycentric and dispersed urban forms prevalent in the contemporary United States.

Altogether, the above evidence suggests that urban sprawl emerges, at least partially, from the numerous interactions between households, firms, governments and other agents. Nonetheless, the extent to which the agents present in contemporary cities self-organize and contribute to outcomes that had not been directly incorporated within the designed plans has been misappreciated by many of the theorists of modern urban and regional planning [12]. It is only during the course of the last three decades that the realization of self-organization in cities has acquired a remarkable momentum.

## 3. Novel Insights into Urban Sprawl from the Complexity Sciences

### 3.1. Landscape Ecology and Fragmentation

Recent decades have witnessed an increasing interest in understanding the relations between the spatial patterns of landscapes and the ecological processes that occur upon them [7]. Urban landscapes can be characterized as a mosaic of land use patches. From this perspective, measuring urban sprawl can benefit from a set of spatial metrics from landscape ecology [40], which serve to quantify two main characteristics of the urban landscape, namely the geometric configuration of patches and their functional composition. From the reviewed definitions, an urban landscape might be considered sprawling when its configuration is irregular and fragmented and its land use composition is segregated [41].

Urbanization throughout the world has happened under very different geographical constraints, historical periods and available technologies, resulting in distinctive spatial signatures. However, despite the apparent complexity and diversity of cities and regions, spatial metrics from landscape ecology suggest that remarkable regularities exist among the spatio-temporal evolution of their land use patterns. In a comparative study of four Chinese cities, Seto and Fragkias [42] determined that, in spite of significant differences on the initial urban structures, economic context and local policies, synergies exist in terms of shape, size and growth rates of land use patches. Similarly, Jenerette and Potere [43] explored a global set of 120 cities and resolved that, while individual cities show continued increases in complexity and fragmentation of land use patches, the inter-city diversity of patterns diminishes, suggesting a tendency towards global urban homogenization. After a thorough comparison of hypotheses regarding the spatio-temporal patterns of urban land use change, Liu et al. [44] determined that, under the contemporary Western socioeconomic context, urbanization globally leads to increasing dispersion of land use, structural fragmentation and shape complexity. Notably, such urban landscape significantly matches many of the connotations of sprawl.

### 3.2. Cellular Automata and Fractal Urban Form

Many of the early studies of urban sprawl were based on models of spatial economics that represented cities as a hierarchy of concentric rings of different land uses centered around a central business district, with the underlying assumption that such spatial distribution corresponds to a robust state of equilibrium between market forces. Nevertheless, in a context of massive migration to cities, technological advances and globalization, the assumption of equilibrium started to appear excessively unrealistic.

These shortcomings were noticed at a time where cellular automata (CA) simulations became one of the prominent approaches to study complex systems within the computational and natural sciences. The standard two-dimensional CA consists of a lattice of cells, which can be in one of the defined possible states (e.g., 'dead' and 'alive') and a set of transition rules. Starting with an initial configuration of cell states, the transition rules iteratively determine the future state of each cell based on the states of its neighboring cells. Following the pioneering foresight from Tobler [45] and Couclelis [46], the seminal work of White and Engelen [47] showed how CA simulations with residential, industrial and commercial states and diverse initial configurations can generate realistic urban patterns at unprecedented spatial resolutions. Similarly, Batty and Xie [48] showcased how CA can successfully replicate a wide variety of urban growth dynamics, from the regular grid of rectangular wards in Savannah, in the American State of Georgia to the emergent suburban sprawl of Buffalo, NY, USA. It did not take long for CA to become one of the most prominent approaches to simulate urban and regional dynamics, and *Environment and Planning B: Planning and Design* devoted an entire special issue to this topic [49].

Cellular models of land use change can be used to explore which kind of rules generate the fragmentation of urban landscapes reviewed above. As cities and regions grow, the patches of urban land uses are expected to grow as well, implying that eventually urban patches coalesce and thus

the small ones disappear. However, this contrasts with the evidence of increasing patch diversity and fragmentation. An explanation might stem from the notion that new urban patches emerge continuously in cities and regions, e.g., residential developments in rural areas, new commercial zones in residential areas and the like. In fact, White and Engelen [47] showed that urban CA models can mimic such behavior by calibrating a stochastic parameter that controls the emergence of new urban patches. Under these circumstances, cellular cities quickly self-organize into a fractal structure, where the size distribution of urban patches exhibits a power law, delineating a landscape with numerous small urban patches and very few large ones. Subsequent examination by White et al. [50] disclosed that such power law scaling holds empirically across a broad range of spatial scales—from the city of Dublin to its greater region, or to the entire country of the Netherlands. While the ubiquity of fractals and power-law scaling in a wide range of complex systems has long been noted by many researchers, their meaning is often unclear. To account for that, White and Engelen [47] refer to several CA investigations which suggest that successful structural evolution of complex systems embedded in variable environments, such as cities, is only possible within fractal configurations, where changes at all scales can be absorbed within the structure of the system. As a matter of fact, the works of Frankhauser [51] and Batty and Longley [52] provide extensive empirical evidence of the fractal organization principles that globally underpin contemporary cities.

The implications of these findings on the spatial structure of complex self-organizing systems such as cities might justify the similarities among the spatio-temporal patterns of urbanization and the global tendency towards fragmentation and sprawl reported above. Nevertheless, the success of the above models in reproducing such behavior might be largely attributed to the fact that the degree of stochasticity can be calibrated to replicate the empirical power-law scaling of patch sizes. Thereupon, in order to understand how the myriad of interactions occurring on a city might prompt its evolution to fragmentation and fractal structures, urban models must integrate more domain knowledge, e.g., in the form of explicit representations of the agents and the socioeconomic principles that guide their behavior.

### 3.3. Integrating Urban Theory and Heterogeneous Agents into Cellular Automata

Recent approaches to embed urban theory and socioeconomic interactions into CA have provided novel insights to the emergence of urban sprawl. The pioneering work of Wu and Webster [53] integrated multi-criteria evaluation into CA simulations in order to assess the profits and externalities of developing vacant cells. A case study of the city of Guangzhou with four simulation schemes representing distinctive regulatory regimes illustrated how different weighting of the criteria lead to different spatial patterns, especially how incentives on the accessibility to city centers and railway stations can help to contend with urban sprawl. Another extension of urban CA models by Yeh and Li [54] introduced the concept of "grey cells" that take continuous values according to its development density (instead of the conventional discrete set of land uses), which allows for simulating urban forms according to the density decay functions of classic location theory. By simulating monocentric and polycentric forms with different levels of density decay in the Pearl River Delta region of China, the authors resolved that the actual development follows a monocentric pattern with slow density decay.

Nevertheless, even when considering polycentric structures, location theory fails to explain the degree of fragmentation encountered in suburban areas. While fragmentation and sprawl might be reproduced through the addition of stochastic perturbation to CA, the fact that cells are fixed in space and restricted to a set of predefined states severely limits their appropriateness to represent human interactions. To overcome these limitations, agent-based models (ABM) have been coupled with CA models of land use in a way that cells delineate the physical landscape and human interactions can be properly represented by individual agents [55]. The work of Irwin and Bockstael [56] developed a model of interactions between rural and residential agents with repelling effects between residential land parcels. Land use change simulations in a suburban area of Maryland showed that the inclusion

of such negative spatial interaction effects generates a landscape that is significantly more fragmented, reproducing more closely the observed patterns of urban sprawl. Coupling a similar ABM with spatial metrics from landscape ecology, Parker and Meretsky [57] resolved that the aversion of urban agents to be located close to other urban agents leads to suburban fragmentation, consistent with existing definitions of urban sprawl. A more thorough ABM of residential interactions by Caruso et al. [58] employed a bidding system based on income, rent prices, accessibility to the city center and residential preferences, and coupled it with a CA in order to simulate the conversion from agricultural to residential land use. Experimentation on the Brussels area showed that such approach replicates the spatial residential dynamics better than classic location theory, as reflected by an analysis of the fragmentation and fractal characteristics of the landscape. In fact, the simulations became quantitatively closer to the empirical land use patterns when buyers prefer proximity to green spaces than to economic amenities, supporting the long-standing view that such residential preferences are among the main drivers of sprawl. Further investigations by Brown and Robinson [59] linked an ABM to a real survey of residential preferences in the Detroit area and determined that increasing heterogeneity among agent's preferences leads to more fragmented patterns and sprawl. Their simulations suggest that, as heterogeneous agents select locations on the basis of variable preferences, they tend to spread themselves out, achieving a higher overall level of satisfaction as reflected in the individual utility functions.

Altogether, these studies suggest that fragmentation and sprawl partially result from negative spatial interaction between residential agents, which seem accentuated as agents become more heterogeneous. Nonetheless, there still remains considerable scope for more integration of socioeconomic principles into the agent's interactions and CA transition rules. In view of the continuous growth of computing power and the increasing availability of disaggregate data, such models hold unprecedented potential to test hypotheses and obtain further insights into urban spatial dynamics and the emergence of sprawl.

## 4. Complexity and Planning

The realization of emergence and self-organization in contemporary cities might be central to understanding the shortcomings of physical land use planning and the corresponding failure of many attempts to contend with urban sprawl reported above [12]. In contemporary cities, urban form emerges not only from the decisions of urban planners and policy-makers but also from the locational choices of residences, firms, real estate developers and other agents that pursue their own interests. This is not to say that planning is futile, since planning instruments can indeed mediate and exert significant influence on how such interactions take place, but rather urges for the development of planning practices that embrace the evolving and self-organizing reality of contemporary cities [60]. In this respect, the sprawl indices from the urban economics perspective offer little practical use to urban planners and policy-makers. As expressed by Song and Knaap [61], "should public officials in Houston, for example, be concerned or pleased that it ranked high in clustering and low in nuclearity? If so, how should they respond?" (p. 213). The same might be said about the literature from the complexity sciences reviewed above, notwithstanding the insights into the emergence of urban sprawl and fragmentation that they provide. Fortunately, recent studies have proposed ways in which approaches form the complexity sciences might be exploited to spatially inform planning.

### 4.1. New Urbanism, Central Places and Fractal Planning

Residential preferences for suburban environments are central to the emergence of urban sprawl. During the course of the contemporary era, notable efforts to manage sprawl such as the "New Urbanism" movement have suggested polycentric and hierarchical urban designs as alternatives to compact and monocentric cities, so that a variety of housing choices is offered while the presence of nearby centers ensures proper access to facilities for daily needs.

Recently, a fractal approach to urban planning has been proposed by Yamu and Frankhauser [62] in order to adapt the principles of the foregoing planning movements to the notion of cities as complex self-organizing systems [63]. The advances that fractal planning represent with respect to previous polycentric approaches are manifold. On the one hand, unlike in central place theory, urban development does not need to be uniformly distributed in space, and natural and environmental constraints can be explicitly considered. In fact, the model has been implemented within a GIS [64] and has been applied to manage sprawl and improve the accessibility to shops, urban amenities and open space in the urban agglomeration of Besançon [65], the Vienna–Bratislava metropolitan region [62], as well as to explore forms that preserve the connectivity of ecological habitats in Besançon [66]. On the other hand, fractal structures have the key property that their border is over-proportionally lengthened with respect to their area. Fractal geometry can therefore be exploited in order to satisfy the residential demand for green environments while explicitly considering planning objectives such as protecting natural habitats and improving accessibility to urban and natural amenities. In this regard, unlike the "New Urbanism", which focuses on the neighborhood scale, the fractal approach has the central advantage of iteratively operating at the regional, urban and neighborhood scale, allowing the multiple socioeconomic and environmental criteria to be assessed properly.

## 4.2. Cellular Simulations to Assess Planning Scenarios

Increasing evidence of planning failures, combined with the realization of emergence and self-organization in contemporary cities, has raised wariness of hazardous outcomes among practitioners. Accordingly, academics and planners are increasingly turning to computational models as a means to simulate future scenarios and virtually explore potential effects that interventions might have from a given starting situation.

A remarkable line of research has been devoted to the application of CA and ABM models to real cities in order to simulate future scenarios. A popular CA framework is the Monitoring Land Use/Cover Dynamics (MOLAND) urban and regional model [67], which integrates economic, demographic, land use and transportation models that operate not only at the cellular level but also at the county and regional level. In an application to the Greater Dublin Region, Barredo et al. [68] determined that, while the observed and simulated patterns show notable structural similarity (as characterized by fractal geometry), comparison matrices and kappa statistics reveal significant differences at the level of individual cells. The latter is due to the fact that MOLAND incorporates stochastic perturbation that ensures that every simulation run produces a different output. Acknowledging the relevance of the stochasticity, Shahumyan et al. [69] employed multiple simulation runs to build probability maps of the potential outcomes of a set of different scenarios for the Greater Dublin Region. The results show that most areas are relatively predictable, which denotes their suitability to a particular land use (e.g., due to natural characteristics or accessibility criteria). Nonetheless, the land use of some areas varies largely among scenarios and simulation runs. In this sense, the simulations can serve to spatially map such areas and explore which kind of interventions are likely to lead to the desired outcome. In another MOLAND application to the Greater Dublin Region, Van de Voorde et al. [70] employed landscape metrics for the calibration and evaluation of future scenarios. Their results suggest that, in contraposition to other scenario definitions that control urban expansion, the "business as usual" scenario shows that most characteristics of urban sprawl, namely fragmentation and increasing shape complexity. Similar approaches have also used MOLAND for the formulation of different scenarios in the Algarve region in Portugal [71], as well as the Flanders state in Belgium [50].

Another of the most widely-used urban CA models is SLEUTH (Slope, Land Cover, Exclusions, Urban Areas, Transportation, Hydrologic) [72], which simulates urban growth according to parameters that control the degree of diffusive growth, outward spread, creation of new centers, and the influence of roads. A distinguishing feature of SLEUTH is the self-modification of the parameters as the simulation evolves, which include a transition towards more road-oriented growth as the road network

develops, or a higher propensity to develop cells with high slope values as available land becomes scarce. This allows for the representation of a second-order self-organization with can be further helpful to reflect changing lifestyles, e.g., changing propensity to choose a house in the suburbs instead of an apartment in the city center. Furthermore, a notable contribution by Silva et al. [73] coupled SLEUTH with an additional CA that allocates the cells that SLEUTH intends to develop in a way that predefined landscape planning goals are met. Such goals are determined according to the values of a set of landscape spatial metrics, which can be prescribed to reflect planning strategies aimed at protecting natural habitats and ecosystems. Extensive experimentation with the metropolitan areas of Lisbon and Porto showed how adopting protective strategies can lead to the preservation of large natural patches and corridors between them. Nevertheless, a shortcoming of SLEUTH might be that relating agents and socioeconomic factors to the parameters that control the different types of urban growth is not straightforward [50]. From the perspective of the explanatory potential of CA and ABM, it seems easier to interpret land use change as result of socioeconomic factors rather than relating it to a particular set of predefined urban growth rules.

Overall, computational simulations hold much potential to spatially inform urban planning by locating areas where urban development is most likely to happen and exploring different scenarios that attempt to reach different planning goals. To this latter end, the central advantage of urban CA and ABM with respect to other black-box approaches (e.g., machine learning algorithms) is their ability to explicitly represent socioeconomic principles into their interactions and transition rules, which allows them to not only reproduce the observed patterns but also to explore alternative forms that might emerge when such interactions are mediated by different planning strategies.

## 5. Discussion

After an extensive review of the literature on sprawl and the main intellectual traditions of sustainability, Neuman [74] concluded that, in order to assess whether compact forms are more sustainable than sprawl, cities must be understood as a process. The evidence reviewed in the present work suggests that sprawl must be assessed not only as part of a process, but as part of a complex process of self-organization. This idea is not new either, as the seminal book of Jacobs [75] already adverted that cities "happen to be problems in organized complexity" (p. 433), and that "the theorists of conventional modern city planning have consistently mistaken cities as problems of simplicity...and have tried to analyze and treat them thus" (p. 435). This paper started by surveying how, more than half century later, many of the economic perspectives on the urban sprawl debate are still largely set in terms of problems of simplicity, and followed by reviewing recent contributions to the topic that build upon the complexity sciences.

The complexity and diversity of contemporary urban forms are presumably responsible for the complications that many multidimensional indices encounter when attempting to quantify sprawl. Regarding urban sprawl as the outcome of a complex process of self-organization might provide insights on how to measure it, since, as reviewed above, the spatial signature of such processes tends to be fractal. In this respect, spatial metrics from landscape ecology might be more appropriate to quantify the morphological characteristics of urban sprawl, since they have been inherently devised to measure complex and fractal patterns. Furthermore, while multidimensional indices of sprawl are often hard to interpret, a key advantage of spatial metrics is that they are also good predictors of the ecosystem's ability to support important ecosystem functions [76]. Nonetheless, spatial metrics operate at a scale that is too coarse to address important dimensions of sprawl such as accessibility [22]. In order to ensure a thorough assessment of urban sprawl, spatial metrics should therefore be combined with measures of network structure, which quantify the connectivity and configuration of street networks that characterize accessibility [13].

Besides the lack of consensual definitions and measures, the contradicting results reported above might be attributed to further epistemological shortcomings. Following the pioneering contributions of Jacobs [75] and Alexander [77], it is now widely acknowledged that the elements that configure

contemporary cities are interrelated in complex ways. Under these circumstances, positivist methods of causal inference and law-like generalizations—which prevail in many of the reviewed economic approaches to assess the costs of sprawl—require significant assumptions and subjective judgements from the researchers in order to reduce such intricate relations to a tractable set of independent and dependent variables. In sharp contrast, the dominant epistemology of the complexity sciences has been computational models and simulations, which allow for experimenting with the higher-level structures that can emerge from the interactions between the elementary agents with few a priori assumptions of how these should be represented [78]. From this perspective, formulations of urban CA and ABM might serve to generate candidate explanations for observed phenomena. In point of fact, the main case reviewed in this article is that the models that incorporate residential preferences and agent heterogeneity configure a candidate explanation for the fragmentation and urban sprawl observed empirically.

Overall, the interactions between the elementary parts and how they lead to the emergence of regularities and higher-level structures is the central question of the complexity sciences and self-organization. However, besides conventional technical issues, as models of complex self-organizing systems, CA and ABM are also exposed to more profound concerns. While a specific instantiation of such models might be able to predict higher-level regularities such as urban sprawl, there most likely exist alternative model formulations that predict very similar outputs [79]. Given that many of the hallmarks of complexity—such as fractals and power-law scaling—emanate from the study of physical and biological systems, the application of models from the complexity sciences to cities and regions runs the risk of conflating essentially different phenomena, and special caution is urged in order to avoid over-relying on deduction from analogies. The main takeaway is that urban CA and ABM might not be seen as a replacement for domain knowledge, but rather as framework in which well known mechanisms of urban theory might be embedded to test their implications in more realistic settings. On the other hand, given that urban CA and ABM include stochastic components, they are not well suited to provide exact predictions but rather to reproduce generic mechanisms that govern the evolution of cities [80]. Accordingly, multiple stochastic runs might be exploited to provide realizations of the variety of paths that the urban system might follow. From this standpoint, interactions and transition rules with explicit urban theory might serve to simulate the effects of interventions such as taxes or incentives to endorse specific locational behaviors. This is in fact one of the major practical reasons to employ such models, as, when confronted by paths that are considered to be more desirable than others, they permit assessing which policies increase the probability of attaining the most desirable future.

Nevertheless, the importance of stochasticity in urban models is not just a matter of acknowledging unpredictable behaviors. Altogether, the ability of CA to generate complex patterns from simple transition rules is misleading for the study of urban systems, whose interactions present substantial heterogeneity in space and human behaviors [81]. While the meaning of stochasticity at the cell level remains elusive, at the agent level, stochasticity can be represented as heterogeneity between agents, which actually seems to be decisive for the emergence of urban sprawl. From this perspective, the sprawl observed in contemporary Western cities might be viewed as the spatial signature of the interactions among an increasingly diverse and complex society, which has self-organized in response to the changes in life modes and the technological developments that followed the industrial revolution. The spatial signature of the cities to come is still largely unknown, but, in view of the inherent complexity of urban systems, the reviewed approaches provide insightful means to understand changes in urban form as cities self-organize to globalization and increasing protagonism of information technologies.

## 6. Conclusions

The present article reviews the main shortcomings of the urban economics and the regional science perspectives to quantify urban sprawl and evaluate its costs, and surveys how recent approaches to

the study of cities from the complexity sciences might be employed to assess urban sprawl. In view of the complexity of urban forms encountered in contemporary cities, spatial metrics from landscape ecology seem better suited to measure urban sprawl than the aggregate sprawl indices employed in many studies. While adopting spatial metrics to measure urban patterns might be a first step towards more comparable methods, they should be complemented with measures of density and accessibility in order to consider the multiple facets of urban sprawl. On the other hand, cellular automata and agent-based models provide novel insights into how interactions between residential agents can lead to the emergence of urban sprawl. As reviewed in this paper, these approaches have seen significant advances over the recent decades, and notable applications aiming to spatially inform urban planning and decision-making have been proposed in the literature. Overall, the complexity sciences hold promising potential to enlighten urban theory, yet further efforts should be devoted to understand the heterogeneity of the human interactions that drive the spatial evolution of cities.

**Author Contributions:** M.B. conceived the main ideas, conducted the literature review and wrote the manuscript; J.C. and S.J. thoroughly revised the manuscript throughout its entire redaction process.

**Funding:** This research has been supported by the École Polytechnique Fédérale de Lausanne (EPFL).

**Conflicts of Interest:** The authors declare no conflict of interest.

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
