# Peer review of "Addressing Urban Sprawl from the Complexity Sciences"

_urbansci, doi:10.3390/urbansci3020060_

Reviewer 1 Report

This paper presents an interesting look at complexity science in urban planning/geography research, and argues for the use of ABMs and CAs. It requires revision to deserve publication, but I believe the revision is perfectly feasible given the recommendations below. In general, the paper seems better suited as a “communication” paper rather than a “research” article. As a literature review, it needs to be more clearly scoped and motivated, with a better explanation of how the review was formally conducted. Finally, complexity science itself needs to be formally introduced in the paper, and its key terms must be defined. Some detailed comments and recommended revisions follow:

Line 9-10: this is an odd claim to make, especially considering the reference is 23 years old and there have been literally hundreds of studies conducted since then that have shed pretty clear light on this question

 Line 16: for more context, if useful, see also https://doi.org/10.1080/01944360408976371 and https://doi.org/10.1080/01944363.2012.666731

 Line 60: and automobile dependence in many cities' contexts

 Line 65: for more context, if useful, see also https://doi.org/10.1080/17549170801903496 and https://doi.org/10.1073/pnas.1504033112

 Line 144: it has been discussed in the mainstream urban planning/geography literature for 20 to 30 years at this point (in fact, as evidenced by your references to work by Portugali, Batty, and others in the 1990s; this has continued steadily through today)

 This brings me to a key concern that must be resolved: while the article gestures at various parts of the planning literature, it fails to draw a broad conclusion about its current state, point to a research gap, identify a methodology to address that gap, and then present findings. If this paper is meant to be a lit review, then it needs to be comprehensive, to clearly argue for its motivation, and to explain the methods by which the lit was assessed and the review was conducted.

 Line 237: the literature reviewed in this subsection should consider also the recent relevant Urb Sci paper that explores self-organization and sprawl in urban location choice, using an ABM: https://doi.org/10.3390/urbansci2030076

 Line 266: you say “the mere emergence of urban sprawl is in clear defiance to the rationalist notion of modern planning, since planning authorities have rarely attempted to impose such pattern explicitly.” This is incorrect, as the modernist mode of 20th century planning very deliberately produced sprawling, functionally-segregated, automobile dependent land use patterns and urban designs – it was the 2nd and 3rd order effects they didn’t anticipate. From civil engineering to urban planning to architecture, the modernist mode of city building produced sprawl through its land use instruments, single-family zoning policies, road geometry standards, parking requirements, setbacks, FAR requirements, street network designs, etc. See for instance Hall’s Cities of Tomorrow (for an international review) or Southworth and Ben-Joseph’s Street Standards and the Shaping of Towns and Cities (for a region-specific review) for thorough reviews.

Section 4: “Complexity and Planning” needs a formal introduction to and overview of complexity science/theory. Its current absence is a major oversight in this paper. It is the logical frame of the paper (and appears in the title), yet without formally describing the terms and theoretical framework, it remains mere jargon that calls out for unpacking.

 Section 4: in the complexity and planning section, consider also this very relevant article that discusses measuring urban form and design outcomes from an explicit complex systems/complexity theory perspective https://doi.org/10.1057/s41289-018-0072-1

Author Response

First of all, we would like to thank the reviewer for the valuable feedback as well as for the provided references, which will be of great value not only for the present manuscript but also for further articles that we are currently preparing.

Regarding the major remark about the clarification of the scope, motivation, identification of research gaps and proposed methodology to address such gaps, we have:

- extended the last paragraphs of the introduction - namely the outline of the article - and added some keywords to distinguish the kind of papers that are reviewed at each part (namely urban economics/regional sciences in the second section, and complexity sciences in the third and fourth).

- changed the titles/subtitles of the second section so that they are more informative and also more consistent with the terminology of urban economics/regional sciences
- extended the first paragraph of the fourth section (on complexity and planning) in order to make more emphasis in the research gaps

We hope that the scope, motivation, research gaps and how the review was conducted becomes clearer now.

Regarding the other remarks, point-by-point (note that line numbers correspond to those of the first manuscript, namely those stated by the reviewer):

Lines 9-10: we have simply deleted the claim since it was indeed problematic

Line 16: while the two articles are indeed in line with the sentences, we believe that it is best not to cite them there since:

- Song and Knaap's article is a case study of Portland and does not even dig deep in the potential connections between planning policies and the evolution of their measures of urban form - we believe that it is best to cite instead a comparative review of empirical evaluations of growth management policies.
- The article of Echenique et al. is more focused on a virtual forecasting exercise (by means of the MEPLAN LUTI model) - we believe that citations in this sentence should refer to empirical evaluations rather than forecasts.

Line 60: indeed, automobile dependence is recurrent throughout the sprawl literature, and accordingly, we have appended it to the sentence.

Line 65: the multidisciplinary review of Clifton et al. has been of major help to refine the scope (it is cited in the modified outline). Accordingly, section 2.1 is concerned with the definition and quantification of urban sprawl from the economic perspective, and therefore we believe that Barrington-Leigh and Millard-Ball's article falls outside such scope.

Line 144: in order to be more precise, we have changed "very recently" for "during the course of the last three decades"

Line 237: although the Urban Science article is in line with the "generative social science" approach of the ABM models reviewed in section 3.3, we believe that the section should focus on models that intend to formulate "candidate explanations" for a pattern observed empirically, i.e., urban sprawl. We therefore believe that the article in Urban Science does not exactly fall inside such scope, since its central goal is more the exploration of the inherent mechanisms of self-organization present in contemporary cities, and there is no direct comparison with empirical measures of sprawl.

Line 266: indeed, the sentence is incorrect. We have simply deleted it, and we hope that the first two sentences of section 4 already convey the intended message: that urban form in contemporary cities emerges not only from planning precepts but also, at least partly, from interactions between agents (e.g., residents and firms)

Section 4: we hope that the modification of the abstract, including the definitions for "complexity" and "complexity sciences" solves this issue. We believe that since such terms are very relevant to the sections 2 and 3, it was better to formally introduce them in the abstract. The reference by Boeing suggested by the reviewer has been cited in the outline (to define complexity), and also in the discussion (to make clear and explicit that there are other measures of form besides spatial metrics).

Reviewer 2 Report

The issues of "urban sprawl" and "compact cities" are important and there are lots of researches in recent years. However, there were only 5 references reviewed form 2016 to 2019. To consider this is a review type of manuscript, I would like to suggest the authors to review more latest references about "urban sprawl" and "compact cities".

Line 2-3, The latest version of "World Urbanization Prospects" is 2018, please update the information and reference [1].

Line 28, " The paper begins by reviewing the literature on urban sprawl and compact cities", however, I can not find the review of "compact cities" in the content. If the "compact cities" is a main issue, please add a section of the review of "compact cities".

Line 33, "... most studies misappreciate the notion that form emerges as the result of multiple...", please cite the references.

Line 303, what does "MOLAND" mean ?

Author Response

First of all, we would like to thank the reviewer for the valuable feedback.

Regarding the lack of recent references, we have updated the last three paragraphs of the introduction in order to set the scope and motivation of the paper more precisely. Accordingly, sections 2.1 and 2.2 are concerned with approaches from the urban economics/regional sciences perspective to evaluate the morphological characteristics and costs (mainly in terms of transportation and environmental performance) of urban sprawl. We have not found many recent references that provide novel insights into such issues. We have found many case studies regarding the impacts of sprawl in public health, promoting green spaces in compact cities, analysis of remote sensing imagery, and even a special issue in "Landscape and Urban Planning" focusing in urban sprawl and social inequity. However, we find that none of this works fall within the scope of sections 2.1 and 2.2. We have nonetheless included a reference to a recent revision of the "Costs of sprawl" by Ewing and Hamidi (2017), whose introduction summarizes the state of the art of measuring sprawl, where we have not noticed any major differences with respect to the content of sections 2.1 and 2.2.

Regarding the rest of the remarks, point by point (note that line numbers correspond to those of the first manuscript, namely those stated by the reviewer):

Line 2-3: we have updated the reference to the most recent revision

Line 28: although the controversies that have arisen throughout the debate on "urban sprawl" and "compact cities" are central to the review, we beleive that the main issue is not in "urban sprawl" and "compact cities" per se, but in the shortcomings of the methods employed to assess their desirability. Accordingly, and in line with the updates in the introduction noted above which specify the motivation and scope of the paper more clearly, we have dropped the term "compact cities".

Line 33: many references could be cited in this context, but we believe that Portugali's "Self-organization and the city" is the most thorough work for this matter. More precisely, chapter 11 "Planning the Unplannable: Self-Organization and City Planning", where the realization of the shortcomings of the economic approaches to cities are referred to (in its historical context) as "The First Planning Dilemma". In any case, this issue is covered in more detail in section 2.3.

Line 303: we have added the words beyond the MOLAND and SLEUTH acronyms

Round  2

Reviewer 1 Report

I believe that the authors have addressed my suggestions and that the paper is worthy of publication.

Reviewer 2 Report

This manuscript has reached for publication.